

# Mauritian snail shells show evidence of extinct predators

Justin Gerlach[1], F. B. Vincent Florens[2] and Owen L. Griffiths[3]

[1] Peterhouse, University of Cambridge, Cambridge, Cambridgeshire, United Kingdom
[2] Tropical Island Biodiversity, Ecology and Conservation Pole of Research, Faculty of Science, University of Mauritius, Reduit, Mauritius
[3] Bioculture Ltd., Senneville, Riviere des Anguilles, Mauritius

## ABSTRACT

Shells of several species of *Tropidophora* land snails of the volcanic island of Mauritius (Mascarenes, SW Indian Ocean), all dated between ~1426BCE and ~1090 CE predating the earliest confirmed human discovery (1519) and settlement (1638) of the island, have been found with characteristic predatory holes. These are either large, irregular holes on the underside of *T. carinata*, or circular holes 0–9–3.3 mm in diameter, in almost the exact same place in *T. ligata*, *T. icterica* and *T. michaudi*. The former have been suggested to be evidence of predation by the extinct red rail *Aphanapteryx bonasia*, which we consider to be probable. The circular holes have not been reported previously. Examination of these shows them to be very regular in shape and size, to be in the centre of a shallow depression marked by two sets of fine grooves at right angles to one another. These holes were compared to damage caused by predators reported to have 'bored' into shells: *Drillus* elaterid beetles, *Poiretia* spiraxid snails and rathouisiid slugs. The damage is most similar to that caused by rathouisiids and we postulate that the holes were caused by a now extinct predator of that family. The only extant members of the family in the Mascarene islands are too small to be the predators. There is no evidence of such predation in recent shells; this is an extinct interaction between an extirpated predator and its prey.

# INTRODUCTION

The island of Mauritius has only been occupied by humans since 1638 but this has been sufficient to result in a devastating ecological impact due to a combination of hunting, invasive species and habitat destruction (*Cheke & Hume, 2008*). As a result, an unusually large number of species are known to have become extinct on the island, famously including the dodo *Raphus cucullatus* (Linnaeus, 1758). The known extinction rate is 41% for native bird species, 53% for reptiles (*Cheke & Hume, 2008*; *Florens, 2013*) and 34% for native snail species (*Griffiths & Florens, 2006*). The extinction rate in insects is much less well known but can be assumed to be comparable.

The high levels of species extinction will inevitably also have led to extinction of species interactions (*Albert et al., 2021*; *Heinen et al., 2023*). This is most obvious in the case of seed dispersing giant tortoises, and for these surrogate species introductions of other species of tortoise provide some means of restoring lost interactions (*Griffiths et al., 2011*). As these

Corresponding author
Justin Gerlach, jg353@cam.ac.uk

'lost interactions' are behavioural they usually leave no direct evidence, unless they were described by early observers. The exceptions to this are interactions involving hard structures: wood, bones or shells. As these can be durable in suitable preservation conditions they can retain evidence of interactions indefinitely. No notable marks have been observed on wood or bone in Mauritius, but interesting marks have been found on snail shells. In 2006 holes in the large extinct snail *Tropidophora carinata* (Born, 1780) were proposed to have been made by the extinct red rail *Aphanapteryx bonasia* (Sélys-Longchamps, 1848) (Jones in *Griffiths & Florens, 2006*; *Cheke & Hume, 2008*). This snail species is known only from subfossil shells other than the report of an apparently fresh shell collected in the 1870s (*Nevill, 1881*), although the shell is now lost and so the record cannot be verified. Very distinct circular perforations are found in three smaller *Tropidophora* species: *T. icterica* (Sowerby, 1847) (subfossil only), *T. michaudi* (Grateloup, 1841) and *T. ligata* (Muller, 1774) (all now very rare but abundant as old shells). These perforated shells have not been reported previously. The abundance of apparently predator damaged shells in these rare or extinct species and the absence of similar damage in any fresh shells suggests that the predator may also be extinct. These observations prompted a review of the damaged shells in an attempt to determine their cause. This review is presented here.

## MATERIALS AND METHODS

### Mauritian predated shells

Shells with damage typical of rat predation (large gnawed holes on the body whorl or apex) are widely distributed across the island and found in all relatively large snail species. These were disregarded and shells with other forms of damage were ascribed to two different categories of damage (Fig. 1):

1. Smashed body whorl—an irregular hole in the shell, taking up at least half the diameter of the whorl.
2. Drilled shell—small, roughly circular holes in the shell.

Mauritian shells showing potential predator damage were examined from collections made in several localities (Fig. 2), in each case a small number of specimens were examined closely; the full number of specimens is given in Table 1:

1. – *Tropidophora icterica* (two specimens, sample A5693). Subfossil and old dead in sandy soil exposed by clearing, E side of Av Victory, Albion, W Mauritius. S20°12′8.57″, E57°24′16.50″. O Griffiths, July 2021.
2. – *Tropidophora ligata* (five specimens, sample A4054). In sandy excavations, approx. 0.5 m deep, 50 m inland, next to Klondike Hotel, Flic en Flac, W. Mauritius. S20°16′ 20.73″, E57°22′19.75″. O Griffiths, June 2012.
3. – *Tropidophora ligata* (twenty four specimens, sample A3912). Subfossil. In sandy trenches in construction site, next to Clinique Occident, N end of Flic en Flac beach, approx. 250 m from sea. O Griffiths, October 2010.

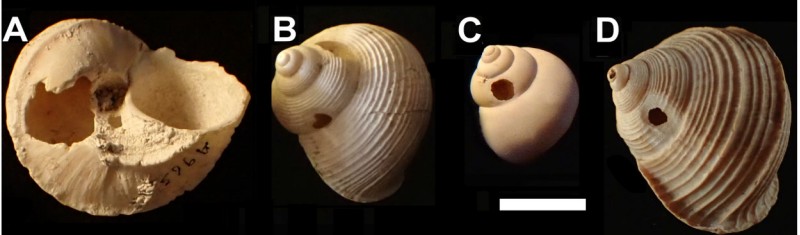

**Figure 1 Shell damage in the different taxa.** (A) *Tropidophora carinata* (smashed body whorl, underside view). (B) *T. icterica* (bored). (C) *T. ligata* (bored). (D) *T. michaudi* (bored). Scale bar 10 mm.          

4. – *Tropidophora michaudi* (six specimens, sample A4106). Deep under piles of rocks in small patch of native dry forest between Riviere Papayes and Rivière du Rempart, just E of village of Cascavelle, Médine SE, W. Mauritius. S20°17′14.07′′, E57°25′16.90″. O Griffiths, January 2013.

5. – *Tropidophora michaudi* (ten specimens, sample A5512). Dead in scree and under rocks in recently exposed bulldozed area within alien vegetation dominated by *Senegalia rugata* and *Hiptage benghalensis*, on east scarp of Rivière du Rempart valley, 500 m SW of Beaux Songes, Mauritius. S20°17′06.70″, E57°25′22.00″. elevation 216–219 m. FBV Florens, C Baider, OL Griffiths et al., March & June 2020.

6. – *Tropidophora michaudi* (four specimens, sample A5587). Under rocks in dry forest at summit of Tourelle du Tamarin, W. Mauritius. Alt 485 m. S20°20′50.65″, E57°22′37.67″. O Griffiths, June 2021.

7. – *Tropidophora carinata* (five specimens, sample A965). In sandy excavations just inland from beach, Riambel, Mauritius. S20°31′01.65″, E57°29′43.40″, OL Griffiths, 1985.

The frequency of damaged and intact shells at these localities and elsewhere was recorded from collected shells (OL Griffiths collection).

Measurements were taken of: each shell (maximum diameter); the holes (height along shell axis and along direction of coiling); their position (number of shell whorls counting from the aperture in the direction of the apex, to the hole; height from shell base; and height from the resting plane of the shell—Fig. 3). All holes were measured using digital callipers, accurate to 0.1 mm.

The edges of the holes were examined microscopically: microscopic grooves were measured and the angle of the grooves measured relative to growth lines in the shells. Angles were measured using a transparent protractor, accurate to one degree. One specimen of *T. ligata* (A3912) was examined using scanning electron microscopy at the Cambridge Imaging Centre.

One specimen from each of the main species was dated at the Cambridge University Department of Earth Sciences Radiocarbon Laboratory using methods of *Freeman et al. (2016)*: *Tropidophora carinata* A965; *T. ligata* A3912; and *T. michaudi* A5512. Approximate sample size varied from 0.47 to 0.87 mgC.

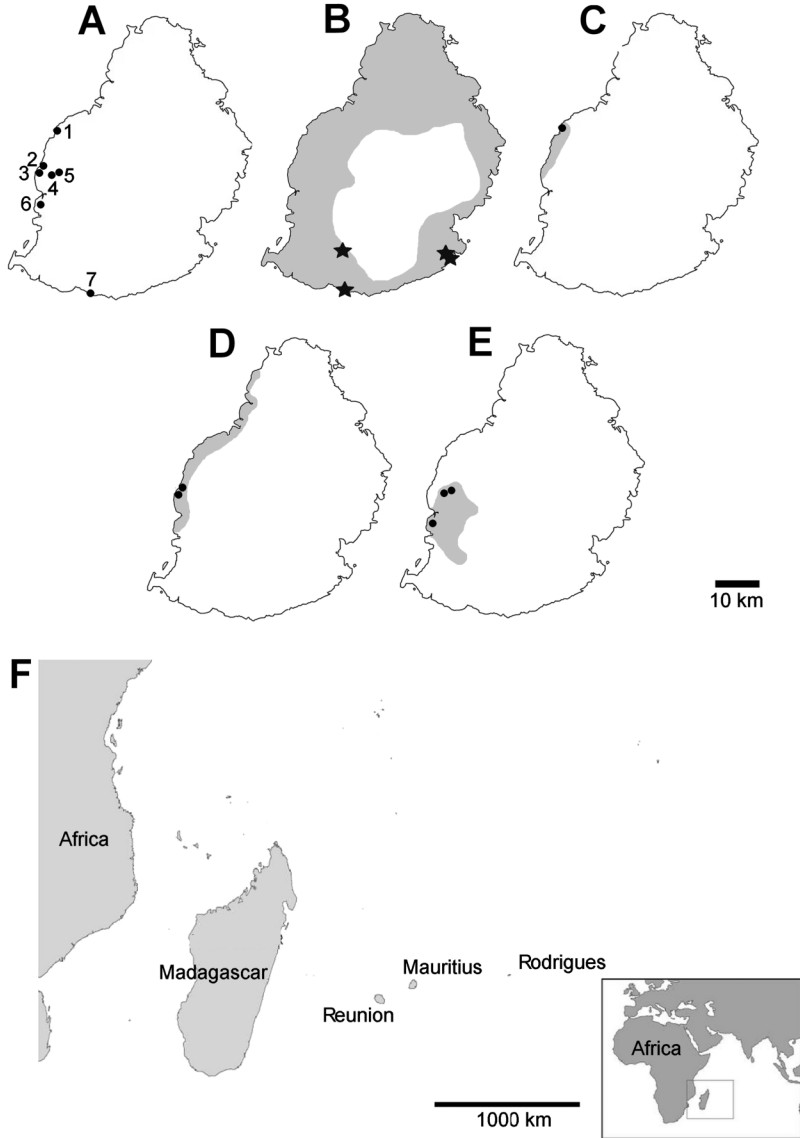

**Figure 2 Localities mentioned in the text, distribution of damaged shells and ranges of the species concerned (distributions from *Griffiths & Florens, 2006*).** (A) localities numbered in the text; (B) *Tropidophora carinata*; (C) *T. icterica*; (D) *T. ligata*; (E) *T. michaudi*; (F) location of the Mascarene islands in the western Indian Ocean. Key to (B–E) shading—approximate original ranges; points—drilled shells; open stars—smashed body whorl; filled stars—drilled and smashed shells. Localities: 1—E side of Av Victory; 2—next to Klondike Hotel; 3—next to Clinique Occident; 4—between Riviere Papayes and Rivière du Rempart; 5—east scarp of Rivière du Rempart; 6—Tourelle du Tamarin; 7—Riambel.

## Distribution of drilling predation marks on *Tropidophora michaudi* shells

To gather further insights on the ecology of the drilled *Tropidophora michaudi*, we examined the distribution of drilled predation marks (both complete and partial) at a site that yielded a particularly large number of shells, sufficient to allow for meaningful

**Table 1 Frequency of damaged shells at different locations showing proportions with the damage considered here and also shells considered to be gnawed by rats.**

| Species | Locality | Sample | % intact or damaged | | | | N |
|---|---|---|---|---|---|---|---|
| | | | Intact | Rail damaged | Drilled | Rat damaged | |
| *T. carinata* | La cambuse | A5544 | 57.1 | 42.9 | 0 | 0 | 28 |
| | Bassin blanc | A1831 | 71.4 | 28.6 | 0 | 0 | 14 |
| | Mare aux songes | A2649 | 61.8 | 38.2 | 0 | 0 | 55 |
| | St Felix | A5574 | 68.2 | 31.8 | 0 | 0 | 22 |
| | Snail rock | – | 41.7 | 0 | 0 | 58.3 | 12 |
| | Riambel | A965 | 71.6 | 28.4 | 0 | 0 | 95 |
| | Nouvelle decouverte cave | – | 100.0 | 0 | 0 | 0 | 40 |
| *T. michaudi* | Cascavelle | A4106 | 60.7 | 0 | 15.4 | 23.9 | 117 |
| | Tourelle du Tamarin | A5587 | 38.9 | 0 | 36.1 | 25.0 | 36 |
| | Flic en flac | A3921 | 59.7 | 0 | 6.0 | 34.3 | 67 |
| *T. icterica* | Albion | A5693 | 86.4 | 0 | 3.0 | 10.6 | 66 |
| *T. ligata* | Flic en flac | A3912 | 70.1 | 0 | 29.9 | 0 | 488 |

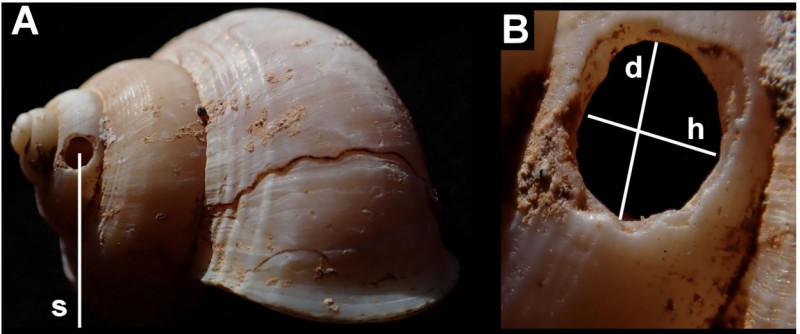

**Figure 3 Shell measurements showing the dimensions recorded in an example of *Tropidophora ligata*.** (A) Shell: s—distance from substrate to centre of hole. (B) Detail of hole, d—diameter in direction of whorl coiling, h—height along shell axis.

statistical testing. The choice of a single site (Rivière du Rempart river valley) also avoided the possible confounding effects that might have arisen from inter-site differences in predation frequency. Of the many individuals found, 230 shells were sufficiently well preserved to allow for both the determination of the snail life stage (adult: with the shell lip expanded; *vs*. non-adult: lacking an expanded lip) along with the detection of presence of successful or attempted shell drilling. We used $\chi^2$ tests to compare the percentages of shells that had completed predation marks (large drilled hole) or attempted predation marks (evidence of initiation but incomplete shell drilling at the usual spot), as well as the percentages of these predation instances on adults *vs*. non-adult shells. We were also able to compare the frequency of drilled predation marks between shells of animals strewn on the ground or under scree at the site, with animals that died during aestivation (as evidenced by their shells being found jammed between sharp edges of rock in the

cavities within a dry layer of undisturbed vesicular volcanic scoria in the ceiling of a thick basaltic rock overhang, where no shell could have fallen from above, but where instead, the snails must have crawled, before dying *in-situ*).

## Comparative predator damage

Predator damaged shells were examined similarly in the Natural History Museum, London (NHM); potential predators were identified by comparison to published descriptions and figures of damage attributed to specific predators (*Schilthuizen, Kemperman & Gittenberger, 1994*; *Baalbergen et al., 2014*; *Helwerda & Schilthuizen, 2014*; *Liew & Schilthuizen, 2014*). Shells apparently predated by *Drilus* sp. beetles (Elateridae) and *Poiretia* sp. snails (Spiraxidae) were found in samples of *Alinda biplicata* (Clausiliidae) (NHMUK 202000133-4), and Rathouisiidae slug bored shells were found in *Plectostoma austeni* (Diplommatinidae) (NHMUK FF Laidlaw colln.).

## RESULTS

### Mauritian shell damage

Shell damage and frequency are summarized in Tables 1, 2 and described below.

1. Smashed body whorl—five *Tropidophora carinata* had irregular holes on the underside of the shell (Fig. 4), next to the aperture or ¾ of a whorl from the aperture. The holes had jagged edges and measured 10.7–13.1 mm high. Width was difficult to determine due to fragmentation in that direction. The shape of the hole suggested punctures caused by an external impact. This damage was found only in *T. carinata* which is the largest of all *Tropidophora* species recorded from Mauritius, and occurs throughout the species' range, except for the very humid central highlands. Damaged shells were found in coastal dunes or in scree. In upland areas shells were only found in caves.

2. Drilled shell—large numbers of shells of *T. icterica* (two), *T. michaudi* (20) and *T. ligata* (28) had almost identical holes (Fig. 4). Holes were almost circular, almost always at least as wide as high (41/50), with a few higher than wide (9/50). Holes were wider than high by up to 44%, but higher by no more than 22%. In *T. michaudi* (and one *T. ligata*) the holes were surrounded by a narrow depression where the substance of the shell had been thinned (Fig. 5, 6). The surface of this depression was covered by irregular parallel grooves measuring 12–29 µm wide. These were in two layers: irregular, deep grooves (Fig. 7E) at 12–26° (19.90 ± 5.09) to the radial sculptural ridges of the shell and more regular, shallow grooves (Figs. 7C, 7G) at 87–110° (95.83 ± 7.44). The deeper grooves were at 83–115° (97.69 ± 9.44) to the shallow grooves. In one *T. michaudi* (A4106) the superficial grooves extended a small distance onto the next whorl (Fig. 8), and in three others (A5512) the eroded and grooved area extended onto two further whorls of the spire (Fig. 8B). This was only superficial but in two cases it extended around approximately half of a whorl (Fig. 8B). In one *T. michaudi* (A5512) and one *T. ligata* (A4054) the depressed eroded area was deepest towards the spire, in other specimens

**Table 2 Mauritian shells examined and measurements of damage.**

| Species | n | Sample | Shell diameter | Hole position | | Hole dimensions (h × d) | Groove width |
|---|---|---|---|---|---|---|---|
| | | | | Whorls | Substrate | | |
| *T. icterica* | 1a | A5693 | 24.3 | 2.0 | 10.8 | 2.0 × 1.3 | – |
| | 1sa | | 18.1 | 1.1 | 6.7 | 1.1 × 1.3 | – |
| *T. michaudi* | 6a | A4106 | 23.3–30.8 | 1.1–1.2 | 6.5–13.2 | 2.3–3.8 × 2.2–4.3 | 15–29 |
| | 4a | A5587 | 18.1–24.0 | 1.1–1.2 | 7.0–7.2 | 1.8–1.9 × 2.1–2.6 | 26 |
| | 5a, 1sa, 4j | A5512 | 12.7–25.6 | (0.1) 1.1–1.2 | 5.8–10.6 | 0.9–3.3 × 1.6–3.3 | 16–25 |
| *T. ligata* | 4a | A4054 | 15.5–18.1 | 1.1 | 6.3–7.3 | 2.5–2.8 × 2.3–3.2 | 22 |
| | 21a, 2sa | A3912 | 15.9–19.9 | 1.0–1.2 | 4.6–9.5 | 2.1–3.1 × 1.9–3.3 | 15–28 |
| *T. carinata* | 5a | A965 | 33.1–34.5 | Under | – | h 10.7–13.1 | – |

**Note:**

Abbreviations: a, adult; d, diameter; h, height; j, juvenile; sa, subadult. Hole position is number of whorls from the aperture (see Fig. 1 where aperture to hole is just over 1 whorl in each case). Distance from substrate in mm, groove widths in μm.

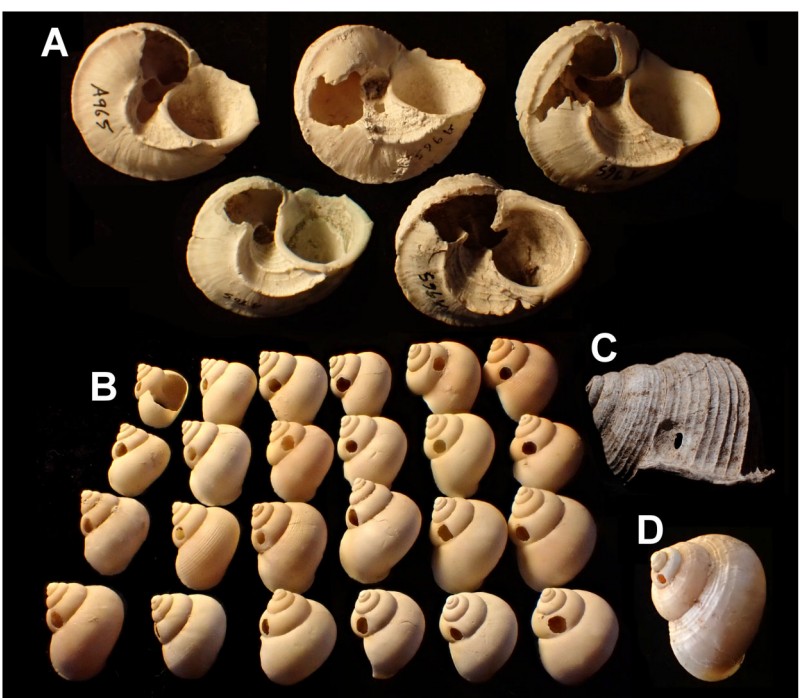

**Figure 4 Variation in predated *Tropidophora*.** (A) Variation in irregular fractured holes in *T. carinata*. (B) Limited variation in drilled holes in *T. ligata*. (C) Exceptional position of a drilled hole in a *T. michaudi*. (D) Exceptional position of a drilled hole in a *T. icterica*. Not to scale.

the depression was regular. Drilled shells come from a restricted area of the prey species' ranges (Fig. 2): throughout the ranges of *T. ligata* (western lowland coastal dry forest) and *T. icterica* (central west coast) in the lowland west of Mauritius, but just the lowland area of *T. michaudi*'s range in the south-west from coast to ridge top.

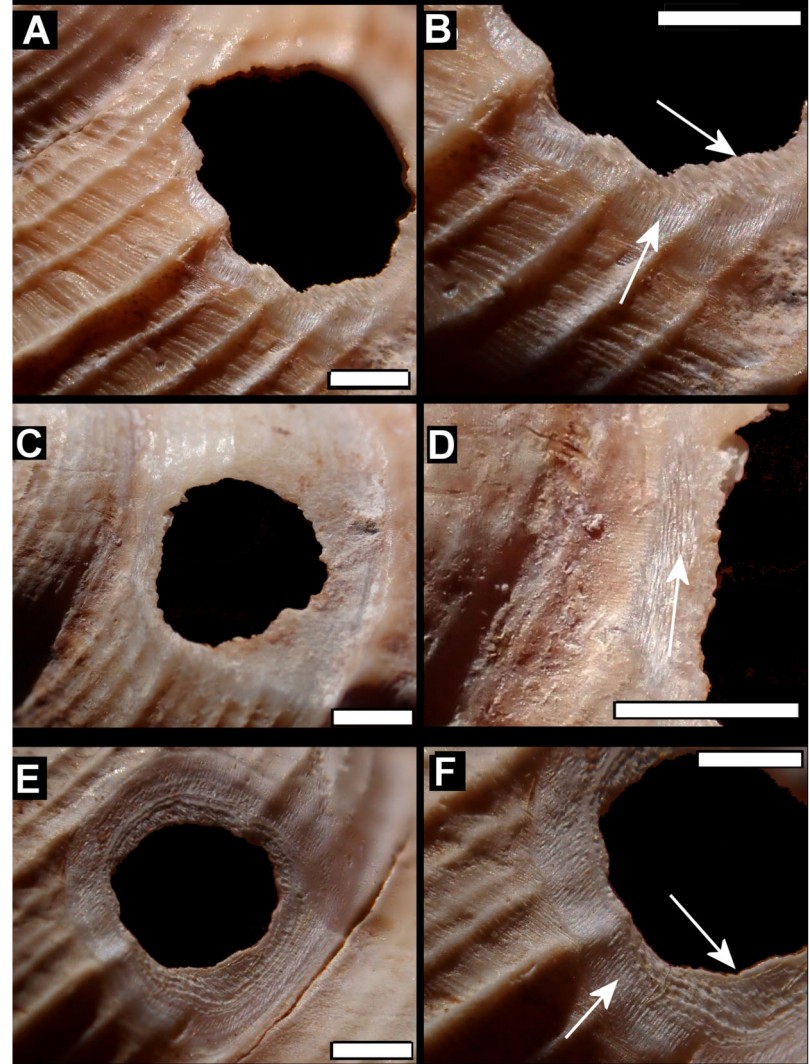

**Figure 5 Details of bored holes in *Tropodiophora michaudi*.** Details showing grooves with shallow grooves denoted by up arrows, deep grooves by down arrow. Scale bar 1.0 mm.

## Dating

All specimens we dated pre-date recorded human occupation of the island by between about half to three millennia (Table 3). These are taken to be indicative of the dates for the shell assemblages, although in at least some cases they may be accumulations over longer time periods.

## Distribution of drilling predation marks on *Tropidophora michaudi* shells

Of the sample of 230 shells of *Tropidophora michaudi* examined from the Rivière du Rempart site, juveniles had sustained significantly higher successful predation (69.2%) than adults (37.5%) ($\chi^2$ = 20.694; df = 1; $P$ = 0.00001). The smallest confirmed juvenile

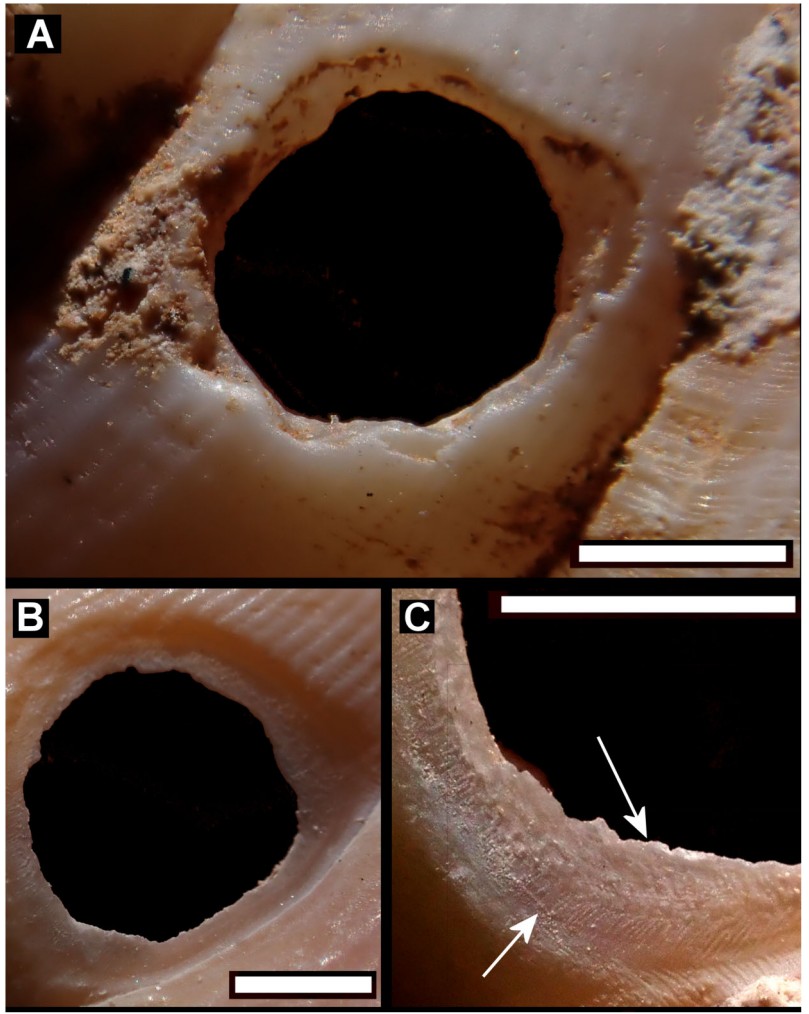

**Figure 6** **Details of bored holes.** (A) *Tropidophora icterica*. (B, C) *T. ligata*: (C) Detail showing shallow grooves denoted by up arrows, deep grooves by down arrow (these are worn and less distinct than the shallow ones). Scale bar 1.0 mm.               

bearing the drilled predation marks was 15.6 mm high, which is between 56-68% of the size of the adult. Among shells that displayed failed predation attempts (initiated, but incomplete shell boring), most were adults (91.7%, $n = 12$) ($\chi^2 = 15.998$; df = 1; $P = 0.00006$). When attempted, the rate of successful predation on adults and juveniles differed significantly and were respectively of 83.8% and 98.2% ($\chi^2 = 7.100$; df = 1; $P = 0.00771$). Of the 14 snails that can be confirmed to have died *in-situ* within their aestivating hideouts, none bore drilled predation marks, whereas predation marks occurred in 111 (48.3%) of the 230 other shells found ($\chi^2 = 12.363$; df = 1; $P = 0.00044$).

## Comparative predator damage

*Poiretia*-made holes in *Alinda biplicata* (Figs. 9A, 9B) measured 1.5–2.0 × 0.8–1.2 mm, taking up most of the height of a whorl, sometimes extending into another whorl. The

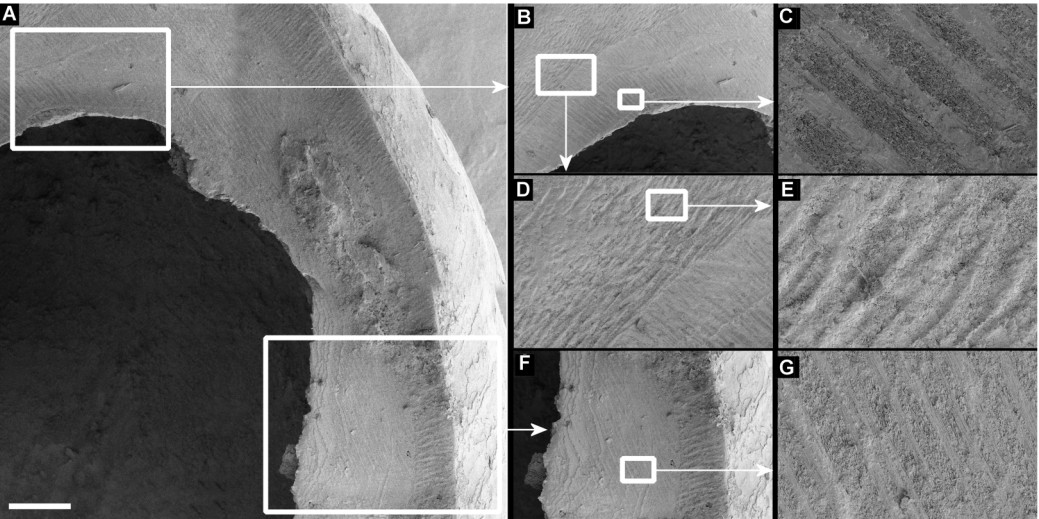

**Figure 7 Scanning electron micrographs of *Tropidophora ligata* (A3912).** (A) Hole showing location of detailed views. (B) Margin of hole showing vertical and horizontal grooves. (C) Detail of shallow grooves. (D) Border between different grooves. (E) Detail of deep grooves. (F) Lower grooved area. (G) Detail of shallow grooves. Scale bar: (A) 325 μm; (B) 400 μm; (C) 30 μm; (D) 100 μm; (E) 30 μm; (F) 300 μm; (G) 30 μm.      

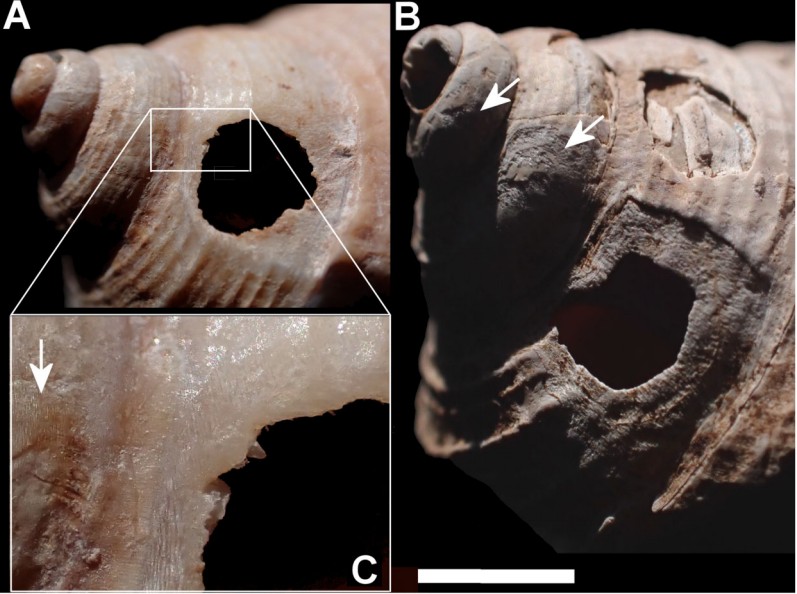

**Figure 8 Superficial grooves (A) and deep gauges (C) extending onto spire in *Tropidophora michaudi* (A4106 and A5512).** Arrows mark the areas of extended abrasion. Scale bar (A) 1 mm (0.3 mm detail), (B) 2 mm.      

**Table 3 Radio-carbon dates of predated Mauritian shells.**

| Species | Sample ID | $^{14}$C Age | $\delta$C$^{13}$ | F$^{14}$C | uAC |
|---|---|---|---|---|---|
| *Tropidophora carinata* | A965/1 | 1,304 ± 26 | −13.5 | 0.8501 ± 0.0028 | 14.1 |
| | A965/2 | 1,216 ± 19 | −12.2 | 0.8595 ± 0.002 | 23.7 |
| *T. ligata* | A3912/1 | 3,254 ± 21 | −5.2 | 0.6669 ± 0.0017 | 21.1 |
| | A3912/2 | 3,363 ± 21 | −11.7 | 0.658 ± 0.0017 | 20.8 |
| *T. michaudi* | A5512/1 | 895 ± 27 | −32.3 | 0.8945 ± 0.003 | 13.8 |
| | A5512/2 | 900 ± 17 | −10 | 0.894 ± 0.0019 | 22 |

**Note:**
$^{14}$C age in years before 1950. $\delta$C$^{13}$—relative deviation of $^{13}$C/$^{12}$C ratio compared to a standard; F$^{14}$C—fraction modern $^{14}$C; uAC—ultraviolet absorbing compounds.

edges of these holes were smooth, lacking notable scratches and surrounded by very extensive etched areas.

*Drilus*-made holes in *Alinda biplicata* (Figs. 9C, 9D) extended across two or three whorls measuring 3.2–4.5 × 1.5–2.0 mm. The edges of these holes were jagged, marked with irregular scratches and had only a small area of etched surface around them.

Rathouisiidae-made holes (Fig. 9E) were minute, sub-circular and with a narrow etched border. The maximum dimension of the hole was 0.4 mm. No other details could be detected on the very small shells concerned.

## DISCUSSION

The shell damage identified here can be ascribed to two different types of predation: large external punctures (*Tropidophora carinata*) and small drilled holes (*T. icterica, T. michaudi* and *T. ligata*). The external punctures appear to have been made by an object about 10 mm wide, which corresponds to the beak diameter of the extinct red rail *Aphanapteryx bonasia* which has been speculated to be the predator (*Griffiths & Florens, 2006*; *Cheke & Hume, 2008*; *Hume, 2017, 2019*). It had the largest beak of any of the Mauritian rails, reaching about 10 cm wide at its midpoint, making it the most likely predator of the family. The Mauritius night heron *Nycticorax mauritianus* (Milne-Edwards 1874) is also a possibility, but no beaks of that species are known and night herons tend to swallow their prey whole rather than to smash it (JP Hume, 2025, personal communication). Red rails were first proposed as snail predators in 1868 (*Milne-Edwards, 1868, 1869a, 1869b*), although as crushing shells in the manner of oystercatchers. The only *T. carinata* populations that lacked evidence of predation were those of the central highlands, outside of the recorded range of the red rail (*Hume, 2019*), although this apparent range is likely to be a preservation bias due to limited calcium availability away from coastal areas. The upland populations of *T. carinata* were also smaller than those from the lowlands (maximum dimension 21 mm compared to 35 mm), making them less attractive prey. Lowland shells are found mainly in dune environments that preserve the thick calcareous shells well (OL Griffiths & FBV Florens, 2025, personal observations). In contrast the upland ones are found in sheltered overhangs where they are protected from the more acidic forest leaf-litter. Thus, the absence of evidence of predation in the uplands

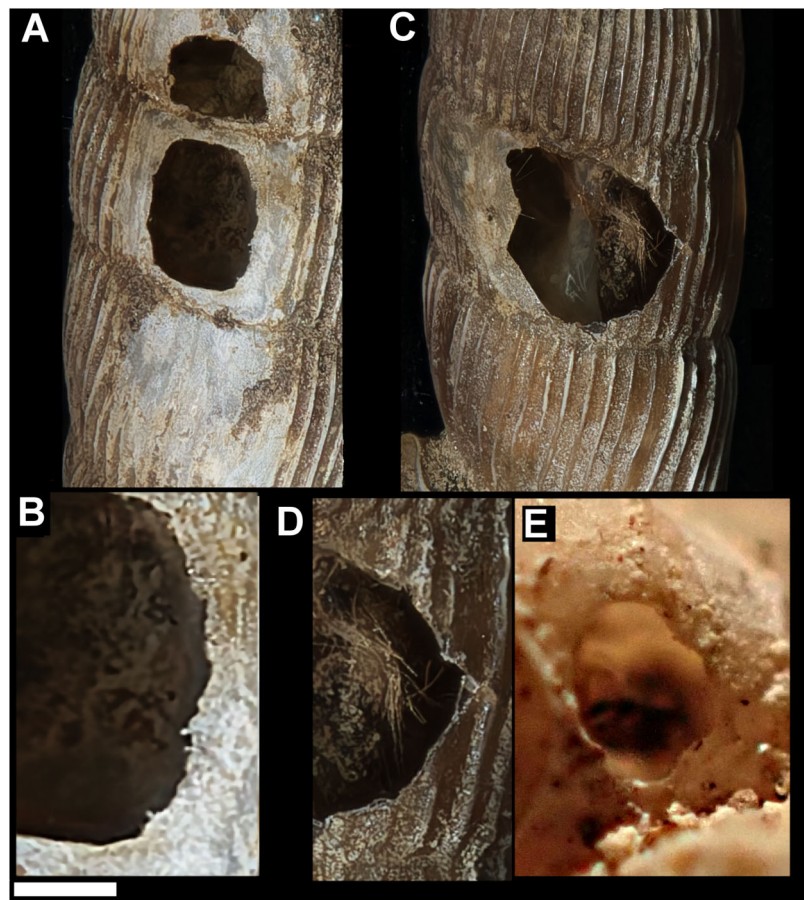

**Figure 9** **Details of comparative predated shells.** (A, B) *Alina biplicata* shells (NHMUK 202000133-4) predated by *Poiretia*; (C, D) same predated by *Drilus*; (E) *Plectostoma austeni* (NHMUK FF Laidlaw colln.) predated by Rathouisiidae, showing broad eroded areas (A) and irregular scrape marks (B) caused by *Poiretia*, and splintered edges (C) and irregular mandibular scratches (D) made by *Drilus*. Scale bar 1 mm (A & C), 0.2 mm (B & D), 0.1 mm (E).

may be due to the absence of predators, avoidance of low value prey or, most probably, poor preservation. Similar damage is caused by the weka *Gallirallus australiae* (*Meads, Walker & Elliott, 1984*) on the thin shells of *Powelliphanta* snails in New Zealand. This is usually by pecking out the spire but high spired shells may be penetrated on the underside (Fig. 10), through the wide umbilicus which "stops the weka's beak from glancing off" (*Meads, Walker & Elliott, 1984*). A similar pattern is seen in shells broken by the Okinawa rail *G. australae* (*Miyazawa & Shimada, 2017*). The positioning of holes in *T. carinata* is comparable. *Meads, Walker & Elliott (1984)* also described predation by New Zealand parrots which made holes 15–30 mm across, "through the outer whorl near the aperture and through all the successively decreasing inner whorls", "on larger, thicker shells there were pairs of vertical scratches around the side of the shell presumably where several attempts were made to penetrate the shell". However, this damage is now known to have been caused by possums (K Walton, 2025, personal communication). The Aldabra rail (*Dryolimnas cuvieri*) has been observed *in-situ* repeatedly hitting an object with the tip of

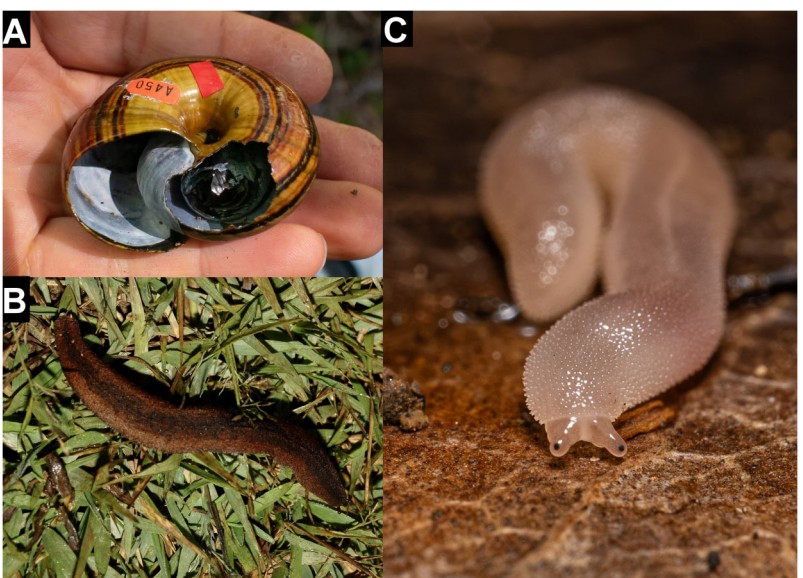

**Figure 10 Comparative photographs.** (A) *Powelliphanta hochstetteri* predated by weka *Galliarallus australiae* (photo: R. Bollongino); (B) Rathouisiidae from Madagascar (photo: L. de Beer); (C) Rathouisiidae from Réunion (photo: N. Huet).               

its closed beak with force, apparently in attempts to crack it open (FBV Florens, 2023, personal observations) in a behaviour reminiscent of what would have cracked the shell of *T. carinata* to create the kind of holes observed on the underside of the shells. The restriction of the holes to the underside of the shell suggests a deliberate turning of the shell to expose the concave surface where such a blow would be most effective. Accordingly, we consider it highly probable that the *T. carinata* shells were predated by red rails.

The drilled holes differed slightly in the different species, being almost vertical sided and usually lacking grooves in *T. ligata* and being always accompanied by a grooved depression in *T. michaudi*. This difference can be explained by the relative thickness of the shells (0.3 mm in *T. ligata*, 0.6 mm in *T. michaudi*). The thicker shells of *T. michaudi* would have been harder to penetrate and require more effort by the predator. Stereotypically positioned holes in shells are known to be produced by human consumption of snails, but these holes are sharp and jagged edged (*Hutterer et al., 2014*), unlike the thinned and smoothed edges of most of the holes in the Mauritian shells. Additionally, the radio-carbon dating shows that the damaged shells predate human occupation of Mauritius by between about half to three millennia. These shells, especially in the case of *T. michaudi* show evidence of the holes being made by a combination of secretion of an acidic substance onto the shell and the scraping away of the softened material to form a small hole.

Acidic secretions are known to be used by *Drilus* elaterid beetles to thin the shells prior to penetration. Entrance holes are small (0.38–0.77 mm) and beveled by acid dissolution, whereas exit holes are vertical without such dissolution, oval and wider: 0.92–1.38 mm (*Orstan, 1999*; *Schilthuizen, Kemperman & Gittenberger, 1994*). These are not generally abraded, and the holes have sharp vertical edges caused by the mandibles

(*Baalbergen et al., 2014*). Carabid ground beetles create distinctive spiral biting patterns running from the shell mouth towards the apex around the spiral (*Němec & Horsák, 2019*; *Millar & Waite, 2004*; *Hayashi & Sugiura, 2021*). Similar damage can be created by opilionids (*Martens, 1965*).

The parallel microscopic grooves in the etched area of the Mauritian bored shells are indicative of rasping by a mollusc radula. Scraped holes of a similar size are known to be caused by Zonitidae such as *Aegopinella nitidula*, the holes formed this way are irregular ovals with a broad worn area around it (*Preece, 1998*; *Myzyk, 2014*). The irregularity of the worn area and the hole, the lack of the etched depression and the varied angles of the scrape marks distinguish zonitid predation from the Mauritian holes. The scraping of the Mauritian holes is very regular, with a superficial scraping at 20° (12–26) to the axis of the shell, followed by rotation through 113° (105–123) to give deeper penetrative gauging at an almost right angle to the axis (96°, 87–110).

Species in four families of molluscs have been identified as secreting acid solutions in predation. The most specialized are the marine Muricidae and Naticidae. Both secrete acid from an accessory boring organ and are able to rotate the odontophore (through 180° in Muricidae and 90° degrees in Naticidae) to make a cylindrical hole with what appears to be a random rasping pattern (*Carriker, 1981*). Of terrestrial species, the Spiraxidae *Poiretia dilatata* secretes acid from a pedal gland over a 2-day period, causing a wide etched area and a large irregular hole. There is no rasping in this species (*Helwerda & Schilthuizen, 2014*; *Helwerda, 2015*). More specialist boring is carried out by Rathouisiidae: *Atopos* spp. produce drill holes that resemble those in Mauritian shells, with a narrow, scraped margin and measuring 0.13–0.33 µm diameter (*Liew & Schilthuizen, 2014*). The drill hole size corresponds to proboscis diameter (*Liew & Schilthuizen, 2014*). *Rathouisia leonina* predating snail eggs made smaller holes up to 0.62 mm in diameter (*Wu et al., 2006*). The presence of an acid etched area, its scoring by parallel grooves and the width of the grooves measuring 12–29 µm all suggest that the predator is likely to be a mollusc, feeding in the same manner as Rathouisiidae. Rathouisiids are known from south-east Asia and Australasia, with undescribed species in Mauritius (*Griffiths & Florens, 2006*) and Réunion (OL Griffiths, 2025, personal observations; Fig. 10). Small slugs are often widely dispersed by human activity, and one rathousiid is known to have been transported this way (*Manganelli et al., 2023*). The Mascarene species do not correspond to any known species and appear to be undescribed. Being restricted to native forest (O Griffiths & V Florens, 2024, personal observations) these may be native species. If so, they would represent something of a biogeographical anomaly, although many Mascarene molluscs have Asian and Pacific origins (*Griffiths & Florens, 2006*). It is worth noting that rathousiids may have been overlooked until recently; there are unpublished observations of the family from North and South America (JF Raupp, 2025, personal communication) and an undescribed large species from Madagascar is shown in Fig. 10. The species known from Mauritius and Réunion are only 12 mm long, so are unlikely to be able to produce large enough holes to be the predator. Asian species are known to reach 85 mm long (*Wiktor, 2003*), so it is possible that a large Mascarene rathousiid may have existed in the past.

Stereotypical positioning of boring holes, as in the present case, is known from naticid whelks (*Berg & Nishenko, 1975*; *Kitchell et al., 1981*; *Archuby & Gordillo, 2018*; *Kingsley-Smith, Richardson & Seed, 2003*) with the position being selected to minimise energy expenditure in drilling (*Mondal, Hutchings & Herbert, 2014*). The stereotypical position of the bored holes could indicate that the predator approached the shell in a stereotypical manner while also taking into consideration the accessibility of the animal when maximally retracted within the shell as a defensive response to the attack. The latter inference is demonstrated by the existence of a few shells that had slightly misplaced drill holes that fell just over the suture onto the body whorl, instead of their characteristic predatorily successful position on the penultimate whorl. These misplaced drill holes were all invariably abandoned as soon as the shell was pierced and therefore as soon as the predator would have sensed the absence of the prey's flesh, leaving only a very small unfinished hole. The predator would then readjust the position of the drilling to create a second and definitive full hole in the correct place, thereby completing the predation event. The fact that exactly the same point on the penultimate whorl was attacked in *T. michaudi* and *T. ligata*, despite the species being of very different sizes, supports the inference that the predator drilled the hole both at a point beyond which the snail could not have retracted in defensive response, and also in a position that is accessible to the predator when the shell is upright on the substratum (incidentally suggesting also that the predator may not have been handling and turning the prey during attack, but instead simply positioned itself on top of it before starting to drill the hole). The holes would be in a consistent position as a predator on top of the snail shell would not tip it over given that the shell would be attached to the substrate, either by the body of the snail or by mucus secreted by the snail to secure the shell at rest.

Orientation of the grooves suggests that the gauging started at the suture with the body whorl and moved towards the spire; grooves start in a straight line at the suture, with no evidence of scraping onto the body whorl (except in the rare instances of originally misplaced drilling mentioned above), whereas at the spire end the groves are irregular. This would indicate that in this stage of feeding the forepart of the predator was oriented along the shell axis, head-up towards the spire, and that the snail was in the normal resting position with the shell's mouth against the stratum. The depth of the depression in two cases indicates that the gauging may have been from the basal end towards the apex, with the deepest part reflecting maximum impact where the rasping came up against the suture. It was not possible to determine the direction of the superficial grooves. From this we deduce that the predator first positioned itself towards the apex of the shell with its head approaching the lip. This would mean that the proximity of the shell's peristome would indicate the boring position. It probably then started scraping at the surface whilst releasing an acidic secretion. The beginning of the shell surface eroding probably then stimulated the predator to change the angle of attack, rotating the odontophore by 90° and gauging along the direction of coil of the shell until a hole was formed. The small number of specimens where the abraded area extended beyond the immediate surroundings of the hole (Fig. 8) may indicate predators with an inefficient technique, either younger animals

or ones affected by different environmental conditions or the position of the prey with the apex pointing down such that some of the secreted acidic substance may have flowed downwards under gravity.

The restriction of the bore holes to just *Tropidophora* species may be explained by the fact that non-operculate species can be attacked faster and with minimal energy expenditure through the aperture, without having to incur the time and energy costs of shell boring. The absence of bore holes on the smaller operculate species (Cyclophoridae, Assimineidae and smaller *Tropidophora* spp.) may have been the consequence of these species being too small to be worth the energetic expenditure of boring when other prey were available (indeed, there exists a large number of non-operculated snail species that could be potential prey and that are smaller than the shells bearing the drilled holes). Of the other Pomatiidae *T. carinata* is probably too thick-shelled for this means of predation, even as small juveniles, but seven other species were large enough and thin-shelled enough to have been potential prey. Three are known from a small number of individuals from single localities, so the lack of predated shells may be uninformative (*T. eugeniae, T. lienardi* and *T. vincentflorensi*). One species, *Cyclotopsis conoidea*, being at most 9.3 mm high, is much smaller than the smallest predated juvenile *T. michaudi* suggesting it would not make an interesting prey for the undescribed predator. The three remaining species (*T. fimbriata, T. mauritiana* and *T. scabra*) all appear to be potential prey in terms of size and geography; the lack of predated shells in these species appears to reflect the abundance of other species of comparable sizes which are non-operculated, and thus easier and less energetically costly to predate.

The analysis of the distribution of drilling predation marks on *Tropidophora michaudi* shells relative to the snail's life stage suggest that juveniles were the preferred prey, presumably because their shells are thinner and therefore faster and less energetically costly to drill through. The advantageous trade-off over a smaller meal that this would represent appears supported by the fact that virtually all failed predation attempts observed (91.7%) had occurred on adult shells and that the vast majority (>98%) of attempted predation on juveniles were successful. It is also notable that we found no evidence of healed holes; attacks either succeeded or failed to penetrate the shell. It appears that predation occurred predominantly if not solely on non-aestivating snails, therefore during the rainier season, suggesting that the predator may have been relatively short-lived, unless it could also feed on other food items to sustain itself during the dry season. Rathousiids are predominantly carnivorous but may also feed on plants and fungi (*Barker, 2001*). Alternatively, the predator also aestivated in the dry season and was active synchronously with the prey.

## CONCLUSIONS

We conclude that the damaged subfossil shells of Mauritian snails preserve evidence of ecological interactions that have been lost with the decline in the snail populations and the probable extinction of the predators. These predators probably included unspecialised shell-smashing predation by the red rail *Aphanapteryx bonasia*, evidence of which is preserved only in the most robust of shells (*Tropidophora carinata*). In contrast, predation

by a mollusc comparable to the Rathouisiidae was highly specialised. The only evidence for the existence of this species is the presence of the highly stereotypical bore holes in *Tropidophora* shells which are half to three millennia old. Given that several new species of molluscs continue to be discovered and described from Mauritius (*Griffiths, 2000*; *Griffiths & Florens, 2004*), it would not be too surprising that a native predatory slug would have existed on Mauritius and gone extinct within the last centuries, leaving no shell behind but leaving predatory marks on its prey, betraying its past existence.

## ACKNOWLEDGEMENTS

We are grateful to Julian Hume for useful discussions and to Jon Ablett for facilitating examination of material in the Natural History Museum, London. We are also grateful to Luke Skinner and the University of Cambridge/14Chrono Centre for the radiocarbon dating of specimens. The SEM images were prepared by Karin Miller at the Cambridge Imaging Centre. We are grateful to Ruth Bollogino for permission to use her photograph of *Powelliphanta* and Len de Beer for use of his of a Madagascan rathouisiid and to Nicholas Huet for his photograph from Réunion. The very constructive reviews helped improve the manuscript.

### Funding

The authors received no funding for this work.

### Competing Interests

Owen Griffiths is a director of Bioculture Ltd.

### Author Contributions

- Justin Gerlach conceived and designed the experiments, performed the experiments, analyzed the data, prepared figures and/or tables, authored or reviewed drafts of the article, and approved the final draft.
- F. B. Vincent Florens conceived and designed the experiments, performed the experiments, authored or reviewed drafts of the article, and approved the final draft.
- Owen L. Griffiths conceived and designed the experiments, performed the experiments, authored or reviewed drafts of the article, and approved the final draft.

### Data Availability

The data are available in the figures and tables.

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
