# Peer review of "Mauritian snail shells show evidence of extinct predators"

_PeerJ, doi:10.7717/peerj.20112_

## Round 0.1 · original submission · Minor Revisions

Thank you for submitting to PeerJ - your paper has received feedback from four experts in the field. They have all indicated that your paper is important and exciting, and offer some suggestions to improve the manuscript.

Please take their comments on board in your revision and address each comment point by point. I look forward to reading your revised manuscript

Reviewer 1 ·

Basic reporting

I agree with the authors' presumption that the snail shell hole was formed by an extinct predator.
The discussion regarding the presumption of extinct predators is somewhat esoteric. Please consider creating a new diagram to aid in your hypothesis. A schematic of how the holes were drilled and the traces left in the holes could be shown.
Since Figure 1 does not have a scale bar, please add the size of the shell in the caption.

Experimental design

This research involves detailed and reproducible experiments, including dating and electron microscopy of the shells.

Validity of the findings

The discussion centers on the estimation of predators. In estimating possible predators, please prepare a table that includes unlikely organisms and summarize the methods used to destroy the shells and the possibility that they were distributed in Mauritius.

·

Basic reporting

I have no comments to make about basic reporting

Experimental design

No comment

Validity of the findings

The conclusions are well stated, linked to the original research question & limited to supporting results.

Additional comments

The paper is well-written and introduces a novel approach to snail predation by a now-extinct, soft-bodied predator that would otherwise remain unknown. Furthermore, the authors, for the first time, have succinctly provided evidence of a now extinct rail predating an extinct species of snail, which increases our knowledge of predator-prey interactions on an oceanic island

·

Basic reporting

The paper is an interesting addition to knowledge of the pre-human ecosystem to Mauritius, is clearly written and referenced. I have added notes on the pdf (also copied below) where I think there are amendments to be considered, including some inconsistencies between text and tables. Some of the tables and and figures require more explanatory labelling. The basic data is included.

Experimental design

The subject matter is not that of an experiment, hence the formal experimental analysis formula is not relevant. However it contains comparative analysis with other studies on snail predation, and these are described and referenced appropriately. The research question arose from the discovery of drilled holes in subfossil snail shells, and this is explored methodically, and a plausible (though not provable) solution is provided in the context of the original (pre-human) ecology of the island.

Validity of the findings

Overall the paper is well-structured and the analysis sound. However there are a number of issues of greater or lesser significance that I have drawn attention to in comment notes on the pdf, copied below. I was particularly puzzled by the statement in the text that only Tropidophora carinata showed attack by (putative) rails, whereas Table 2 lists a good number also of T.michaudi showing 'rail attack'. This inconsistency must be addressed ! (notes for line 152, Table 2 and fig.2). Another important point is that the authors do not demonstrate that the small undescribed rathousiid snails present on Mauritius (and known for 2 decades) are native/endemic, while implying that they are - if they are there is an unexpected biogeographical anomaly to mention. While these points do not materially alter the results or conclusion, they are nonetheless important enough to detract from the scientific accuracy of the document.

line 53
This suggestion was actually made by Carl Jones in the foreword to Griffiths & Florens's book - so should be cited as 'Jones in G&F' (although also mentioned in a photo caption of T.carinata shells, it does not appear in the book's main text). It was also suggested independently by Cheke & Hume (2008: p.286 and note 83 to Chapter 3), where there is some further discussion on the suitability of the predator and its similarity to other snail specialists.

lines 74-5
the small numbers in this list needs to be explained in relation to the much larger numbers in table 2 - is it that only these small numbers were carefully studied and measured, although similar damage was noted in much larger numbers of shells ?

line 152
Table 2 suggests rail damage was also frequent in T.michaudi shells - the inconsistency needs addressing

line 178
it would be more accurate to state that 'all specimens we carbon-dated pre-date...', and then justify the assumption that this applies to the all samples by their occurrence together - though it seems to me that an accumulation could have occurred over a very long time period in some locations, so that the dating assumption should be qualified.

lines 211-214
add Cheke & Hume 2008; The night heron was probably a lizard predator, but if taking snails would be more likely to smash them on a hard surface; it's bill, presumably similar to congeners, being too large to make neat(ish) holes in the snails.

lines 281-285
It is unfortunate that after nearly 20 years the Mauritian Atopos illustrated in G&F (2006) remain undescribed - indeed can the authors show that these animals are in fact native ? Very small slugs could easily be imported. While the suggestion of an extinct large rathousiid is certainly plausible, I would like to see me evidence that the existing species are native/endemic. I would note further that as rathousiids are Asian/Australasian in distribution, and (in default of much phylogeographic evidence) that most Mascarene snails appear to have Malagasy/African affinities, so that this biogeographical anomaly would be worth a mention.

Table 1
The hole position description should be cross-referenced to the figures, as it is then totally clear what 'number of whorls from the aperture' means.
'Distance from the substrate' needs explanation - is it distance of the ground assuming the snail is upright ? Is this a meaningful statistic when the weight of the predator might surely make the victim fall over ? In any case it need a cross-reference to Fig.3 unless the table and figure appear on the same page (ideal).

Table 2
NB rail damaged T.michaudi not mentioned, and indeed denied, in text.

Table 3
isn't 1950 (not 1958) the usual base date for radio-carbon dates ?
The codes in the table headings need to be explained.

Fig.1.
It would help to add 10mm scales to each snail, as, despite the disclaimer, it looks as if the holes in T.michaudi are much larger than in the other two bored species.

Fig.2
the locations 1-7 in A need to be listed in the caption.
What about the rail-damaged shells of T.michaudi listed in Table 2 ? - why no stars for them ?

Additional comments

Although I think the paper should be accepted with minor revisions, they are minor only in the sense that they do not alter the results or conclusions of the paper - however I would like to review changes before publication.

·

Basic reporting

no comment, although a few references are hope to added (see in additional comments)

Experimental design

no comment

Validity of the findings

Generally no comment. In Fig. 9E, this hole made by Rathouisiidae is hoped to be observed using SEM, by which the photo can be compared to those in Fig. 7 of Tropidophora ligata.

Additional comments

While the paper largely addresses the questions readers would expect to be answered, as a reviewer, I still have several points I would like the authors to address:

1. Line 151 'The shape of the hole suggested a single puncture caused by an external impact': might it be multiple punctures resulting in one hole?
2. Lines 155-156 ' This may reflect a genuine difference in distribution of smashed shells or a preservation bias, with broken shell persistence only when buried.' : This sentence is to be moved to the part Discussion.
3. Line 201 'semi-circular': This shape (Fig. 9e) can hardly be called 'semi-circular', perhaps it is a better fit using 'subcircular'?
4. Lines 218-221 'The upland populations of T. carinata were also smaller than those from the lowlands (maximum dimension 21 mm compared to 35 mm), making them less attractive prey. Lowland shells are found mainly in dune environments that preserve the thick calcareous shells well': References are needed here.
5. Line 234 the example/observation of the Aldabra rail: The examples employed in this paragraph can hardly convince the reader why the hole is on the underside of the snail. The predator, presumably, was able to hold/fix the prey and expose its underside to the predators beak/muzzle.
6. Line 347-350 'It also appears that predation occurred predominantly if not solely on non-aestivating snails, therefore during the rainier season, suggesting that the predator may have been relatively short-lived, unless it could also feed on other food items to sustain itself during the dry season.': another possible explanation might be, the predator may also aestivate synchronously when the prey aestivate during the dry season.
7. Is there any shell repair in Tropidophora observed by the authors in this work? Or, does the attacked snail destine to die? This will be also insterested and probably questioned by the readers of this paper.

Some minor errors, like:
1. A, B, C... in legends of figures should coincide with those in the figures.
2. In legend of Figure 9, 'spame' should be 'same'?
3. The scale bar will be in good shape if they are of the same thickness across the figures.
...

---

## Round 0.2 · Minor Revisions

While I appreciate that you have made changes in the manuscript, the rebuttal letter is not detailed and it is difficult to see where exactly in the manuscript changes have been made and for what reason. You did not quote the edited text to indicate how the changes have been incorporated and did not give line numbers where changes were made. If you had incorporated the feedback requested, the line numbers should have drastically changed from the original to the revision.

Without this explicit information in the rebuttal letter, it is extremely difficult for me to see if and how the manuscript has improved. Please remember that the rebuttal letter is for me (academic editor) to see clearly and point by point that the comments have been adequately addressed. And that there are resulting changes in the manuscript that improve the clarity and accuracy of the work. Reviewers represent your ultimate reader base, and any question that a reviewer has will likely be shared by your readership. Therefore, if something is unclear to a reviewer you should make a point to address it in the main text of the manuscript because it likely means the intended meaning did not make it onto the page.

Additionally, you seem to have only partially addressed comments in the letter, but not made direct changes in the manuscript. For example: reviewer comment about original lines 281-285, and your comment says that this cannot be addressed. Why ? and what is the discussion about this particular point in the text of the manuscript?

There are several comments that were copied into your rebuttal letter but not addressed. For example, a reviewer asked you to clarify what the "'number of whorls from the aperture'" meant. And to address how the date 1958 in table 3 was determined. And the comment by reviewer 2 "Table 2
NB rail damaged T.michaudi not mentioned, and indeed denied, in text." was not included in the rebuttal letter.

Please ensure you are more detailed in your revision, and please make it clear exactly how the manuscript has been adjusted and improved. Because some comments were not addressed, and some missing from your rebuttal letter completely, I am therefore requesting that you copy each comment made by a reviewer in text and from the annotated PDFs into your response/rebuttal letter, and address each one point by point with exactly how the text was adjusted and what line it occurs in. Your reviewer comments were thoughtful and reasonable; therefore, I request that your revision and response be as well.

---

## Round 0.3 · accepted · Accept

Thank you for your clear rebuttal letter and edits to the manuscript. I believe that it is now ready for publication. Congratulations!